# Enhanced superconductivity accompanying a Lifshitz transition in electron-doped FeSe monolayer

X. Shi[1,*], Z.-Q. Han[1,2,*], X.-L. Peng[1], P. Richard[1,3,4], T. Qian[1,3], X.-X. Wu[1], M.-W. Qiu[1], S.C. Wang[2], J.P. Hu[1,3,4], Y.-J. Sun[1] & H. Ding[1,3,4]

The origin of enhanced superconductivity over 50 K in the recently discovered FeSe monolayer films grown on SrTiO$_3$ (STO), as compared to 8 K in bulk FeSe, is intensely debated. As with the ferrochalcogenides A$_x$Fe$_{2-y}$Se$_2$ and potassium-doped FeSe, which also have a relatively high-superconducting critical temperature ($T_c$), the Fermi surface (FS) of the FeSe/STO monolayer films is free of hole-like FS, suggesting that a Lifshitz transition by which these hole FSs vanish may help increasing $T_c$. However, the fundamental reasons explaining this increase of $T_c$ remain unclear. Here we report a 15 K jump of $T_c$ accompanying a second Lifshitz transition characterized by the emergence of an electron pocket at the Brillouin zone centre, which is triggered by high-electron doping following *in situ* deposition of potassium on FeSe/STO monolayer films. Our results suggest that the pairing interactions are orbital dependent in generating enhanced superconductivity in FeSe.

[1] Beijing National Laboratory for Condensed Matter Physics and Institute of Physics, Chinese Academy of Sciences, Beijing 100190, China. [2] Department of Physics, Beijing Key Laboratory of Opto-Electronic Functional Materials and Micro-nano Devices, Renmin University of China, Beijing 100872, China. [3] Collaborative Innovation Center of Quantum Matter, Beijing 100190, China. [4] School of Physical Sciences, University of Chinese Academy of Sciences, Beijing 100190, China. * These authors contributed equally to this work. Correspondence and requests for materials should be addressed to Y.-J.S. (email: yjsun@iphy.ac.cn) or to H.D. (email: dingh@iphy.ac.cn).

Until now, the highest $T_c$ among all iron-based super-conductors is achieved in FeSe monolayer films[1–6]. The exact mechanism of this superconductivity enhancement in these systems, as well as in other FeSe-based materials such as $A_xFe_{2-y}Se_2$ (ref. 7) and (Li,Fe)OHFeSe (ref. 8), has become a central focus in iron-based superconductivity. FeSe-based materials with relatively high $T_c'$s share one common key point in their Fermi surface (FS) topology: the absence of hole pockets at the Brillouin zone (BZ) centre[9–11]. The importance of this FS topology to superconductivity has been further supported by doping electron carriers on the surface of FeSe films or crystals using potassium deposition[12–14], or in their bulk using liquid-gating technique[15]. In this context, it is natural to ask up to what level the monolayer FeSe/STO can be electrondoped and how superconductivity is linked to the fermiology at high electron doping.

In this paper we perform angle-resolved photoemission spectroscopy (ARPES) measurements on electron-doped FeSe/STO monolayers. Starting from a sample annealed at 350 °C for 20 h, which already transfers a relatively high-electron concentration into the samples, as confirmed by large electron FS pockets, we deposit K atoms *in situ* onto the surface and achieve a higher doping level.

## Results

**K deposition and FS evolution.** As determined in previous ARPES studies[2,3] and in our current experiment, the FS topology of FeSe/STO(001) monolayer films shown in Fig. 1a consists of nearly doubly degenerate electron-like pockets centred at the M point, in contrast to FeSe bulk crystals[16] and most of the ferropnictide superconductors[17]. We then deposit potassium (K) onto the surface of the film and check the evolution of the FS. Figure 1b shows the FS map after evaporating a small dose of K (as defined in Supplementary Fig. 1). The area of electron pocket at M increases from 8.2% ± 0.2% of the BZ in the pristine sample (Fig. 1a) to 10.4% ± 0.2%, indicating that K atoms introduce extra electron carriers into the system. According to the Luttinger theorem, which states that the electron concentration is given by the area of the FS, we deduce that the electron concentration $x$ increases from 0.164 ± 0.004 to 0.208 ± 0.004. However, further

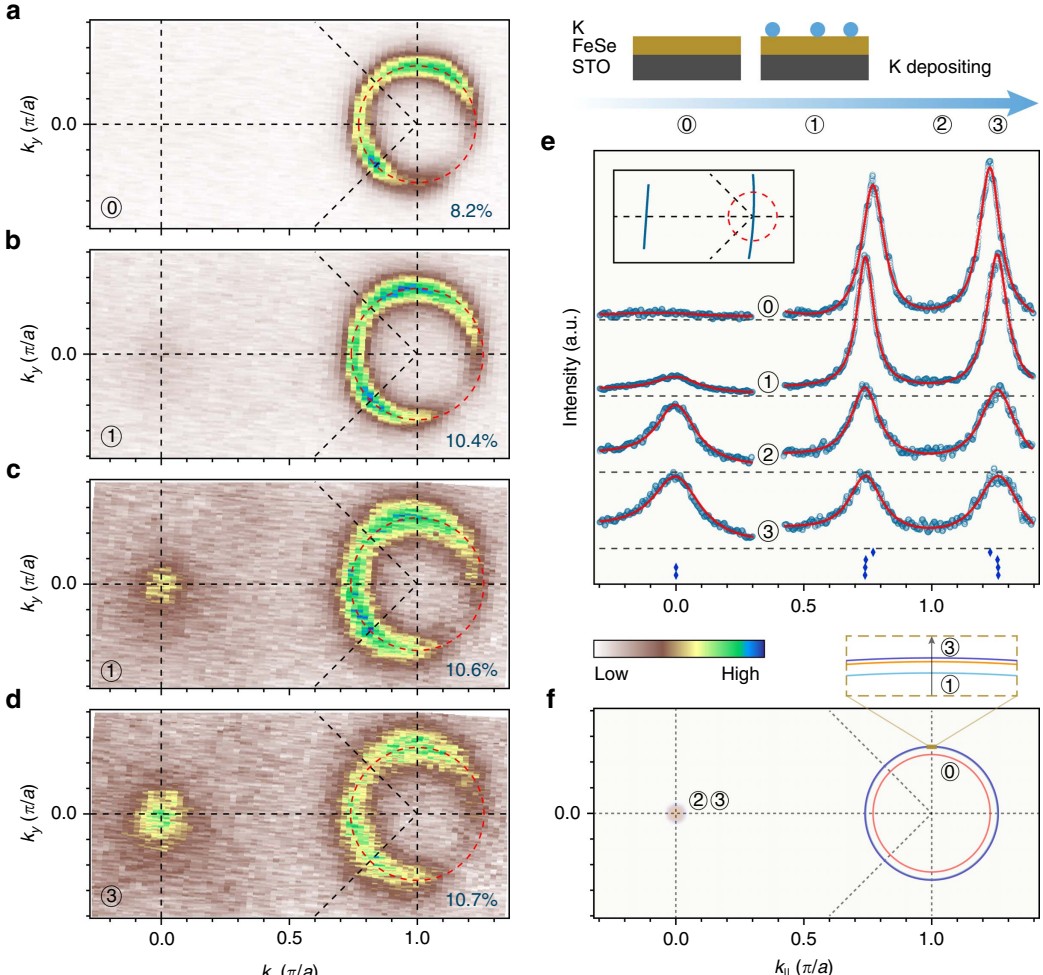

**Figure 1 | FS evolution of potassium-coated 1UC FeSe/STO.** (**a**) FS intensity map of a pristine sample recorded at 20 K and integrated within a 20 meV energy window with respect to $E_F$. The doubly degenerate electron pocket occupies an area of ∼8.2% of the whole BZ, thus giving an electron carrier concentration of 0.164 electrons per unit cell according to the Luttinger theorem. (**b–d**) Same as **a** but for the film after potassium was coated continuously. The percentages at the right bottom of each panel indicate the areas of the electron FS around M in the 1-Fe BZ. The size of the electron pocket at M is saturating slowly upon successive rounds of potassium deposition (∼10.4%, ∼10.6% and ∼10.7% for the first, second and third rounds of deposition, respectively), but instead the spectrum gets broadened, due to the induced disorder at the surface. (**e**) Evolution of the momentum distribution curves along the high-symmetry cuts indicated in the inset upon potassium coating. The red curves correspond to fits of the data using multiple Lorentz functions. (**f**) Comparison of the FSs shown in **a–d**.

deposition of a similar dose of K does not introduce as many electrons as the first time, and the electron concentration of the system slowly saturates ($0.214 \pm 0.005$ after the third round of K deposition), as shown in Fig. 1c,d and Supplementary Fig. 1.

Surprisingly, the FSs obtained after more than one round of deposition exhibit strong intensity at $\Gamma$. This is clearly confirmed by the momentum distribution curves shown in Fig. 1e. Such a change suggests that the system evolves towards a Lifshitz transition, possibly caused by a chemical potential shift. Figure 1f plots the FS evolution of the monolayer FeSe/STO upon K coating, which is more complicated than for the reported results on FeSe thick film[12,13].

**Emergence of electron band at $\Gamma$.** To understand where the intensity at $\Gamma$ originates from, we investigate carefully the low-energy electronic structure in detail. We show in Fig. 2a,b the band structure near $\Gamma$ and M, respectively, along the cuts indicated in the inset of Fig. 1f. The hole-like bands around $\Gamma$ and the electron-like bands around M shift toward high-binding energy, which is consistent with the expected electron doping by K atoms. We note that a simple rigid chemical potential shift cannot describe the band structure evolution, like in the case of K-doped FeSe thick films[13]. Figure 2c shows the spectra recorded at 70 K after division by the Fermi–Dirac (FD) function convoluted by a Gaussian resolution function. While no band is

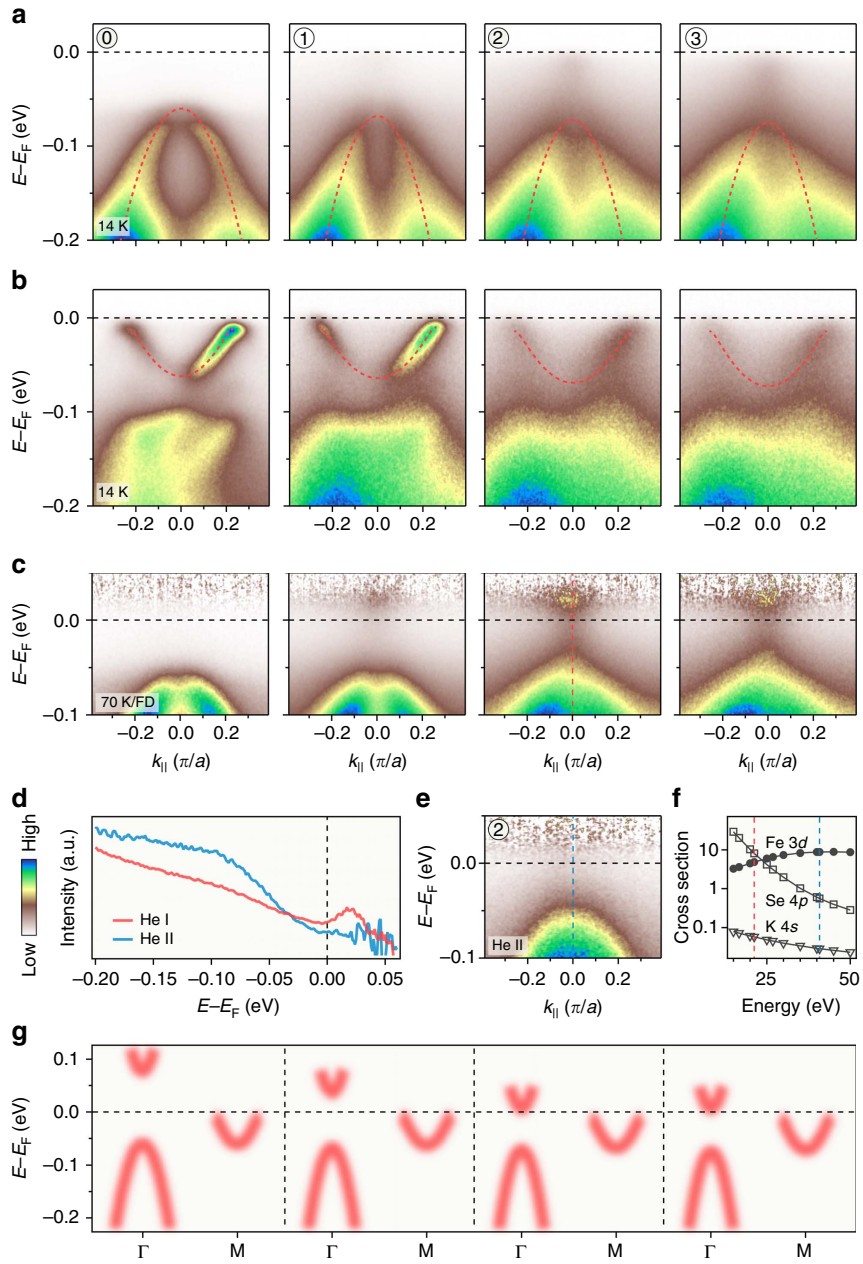

**Figure 2 | Electronic band structure.** (**a,b**) Potassium coating evolution of the ARPES intensity plots at 14 K near $\Gamma$ and M along the direction shown in the inset of Fig. 1e. The dashed red curves are parabolic fits to the band dispersions. (**c**) Intensity plots along the same cut as in **a**, but recorded at 70 K. The plots are divided by the Fermi–Dirac distribution function convoluted by the resolution function to visualize the states above $E_F$. (**e**) Intensity plot near $\Gamma$ recorded with He II rather than He I$\alpha$ photons for the potassium coated sample labelled as 2. (**d**) Comparison of the EDCs at $\Gamma$ recorded with He I$\alpha$ and He II beams. (**f**) Calculated atomic photonionization cross sections for Fe 3d, Se 4p and K 4s. (**g**) Comparison of the band dispersions along the $\Gamma$–M high-symmetry line. The energy positions of the electron-like band around $\Gamma$ are taken from the reference or estimated from the data in **c**.

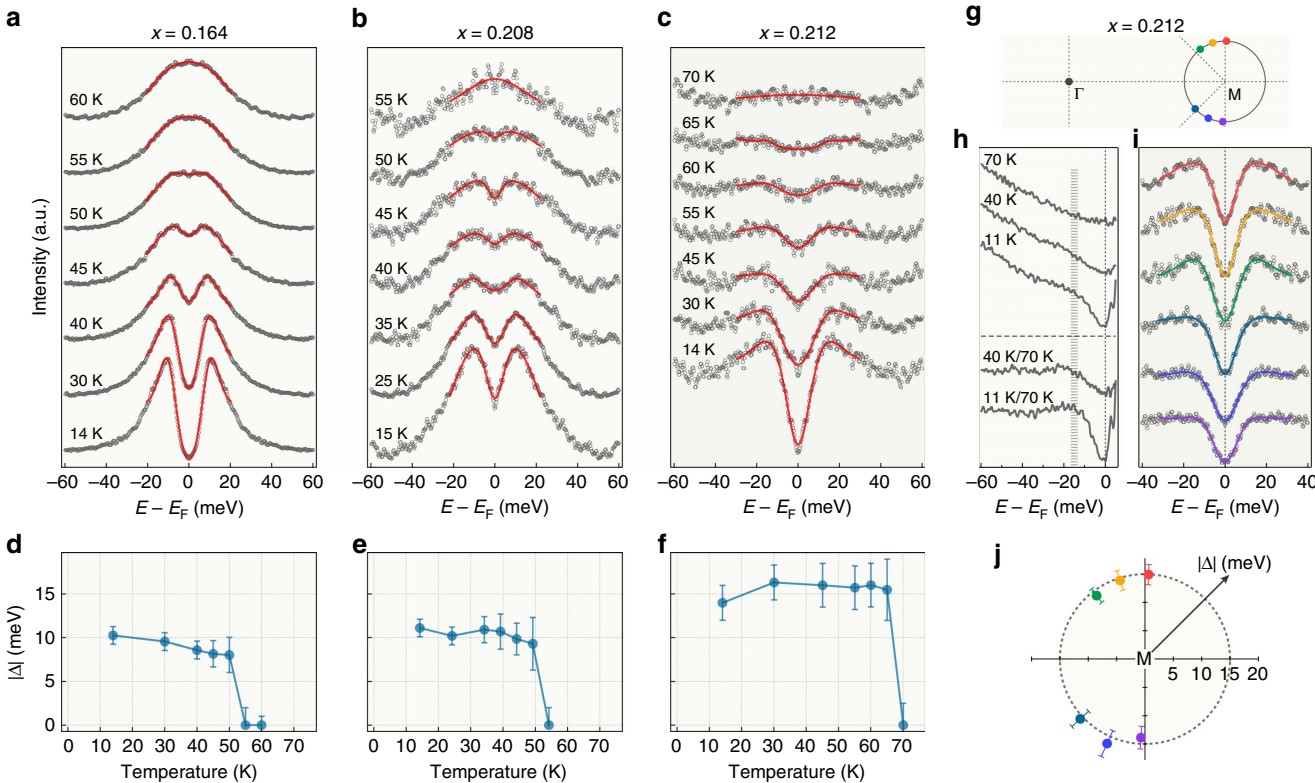

**Figure 3 | Superconducting gap.** (**a**–**c**) Temperature evolution of the symmetrized EDCs at the $k_F$ point of the electron FS around M for pristine and potassium-coated 1UC FeSe/STO. The electron doping is indicated above the panels. The red curves correspond to fit of the data. (**d**–**f**) Superconducting gap sizes as a function of temperature obtained from the fits shown in **a**–**c**, respectively. Error bars are estimated from the standard deviation (s.d.) of the fitting. (**g**) Schematic FS of K-coated 1UC FeSe/STO with doping $x \sim 0.212$. (**h**) Upper part: temperature evolution of the EDCs at $\Gamma$ divided by the FD function. Lower part: low temperature EDCs further divided by the one at 70 K. (**i**) Symmetrized EDCs at 14 K measured at various $k_F$ points as indicated by coloured dots in **g**. (**j**) Polar representation of the momentum dependence of the superconducting gap size for the electron FS around M. Error bars again are estimated from s.d. of the fitting. A nearly isotropic gap is highlighted by the dashed grey circle at 15 meV.

observed in the pristine sample in the measured range above the Fermi level ($E_F$) at $\Gamma$, an additional electron-like band possibly crossing $E_F$ appears for the $x = 0.212$ and 0.214 samples on the unoccupied side of the spectrum (see also the EDCs in Supplementary Fig. 2c). Scanning tunnelling microscopy measurements reveal that the bottom of this band locates at 75 meV above $E_F$ in the pristine monolayer[18]. By comparing results using the He Iα (21.218 eV) and He II (40.814 eV) lines of a He discharge lamp (Fig. 2d), and taking advantage of the opposite behaviour of the photoemission cross sections of Fe 3d and Se 4p in this energy range (Fig. 2f)[19], we conclude that this electron band has a dominant Se 4p orbital character. We note that K 4s states have an even much smaller cross section in this energy range and could be hardly observed, thus excluding the possibility of a K impurity band. Band calculations[18,20] and previous ARPES studies on similar materials[21] demonstrate that the Se $4p_z$ orbital is hybridized with the Fe $3d_{xy}$ orbital at $\Gamma$. It is known that the position of this band is quite sensitive to the Se height (on the very top of the film)[18,20]. In fact, our calculations (Supplementary Fig. 3) demonstrate that K deposition causes this band to shift down and leads to the decrease of the gap between the $d_{xy}/p_z$ band and the $d_{xz}/d_{yz}$ band at $\Gamma$, which is consistent with the experimental data.

**Enhanced superconductivity**. We then check the super-conductivity of the samples. Following a standard procedure[22], we show in Fig. 3a the temperature dependence of the symmetrized EDCs at $k_F$ near the M point for the pristine

FeSe/STO monolayer. In agreement with previous ARPES results[2,3,23,24], the FS is clearly gapped at low temperatures. We fit the experimental data with a phenomenological model for the superconducting gap[25], and display the extracted results in Fig. 3d. The gap size is about 10 meV and closes at around 55 K, which is comparable to reported values[2,3,23,24]. Similarly, we show symmetrized EDCs in Fig. 3b,c for electron doping levels $x = 0.208$ and 0.212, respectively, after K deposition. The corresponding fitting results are displayed in Fig. 3e,f. For $x = 0.208$, the gap size and the $T_c$ does not change much. Interestingly, the gap size jumps to $15 \pm 1.5$ meV and the closing temperature increases to $70 \pm 5$ K after further K doping to $x = 0.212$. We have checked that these values are unchanged within the experimental uncertainties upon further doping, as shown in Supplementary Fig. 4.

The gap that we observe is symmetric with respect to $E_F$ in all the low temperature spectra (Supplementary Fig. 5), which is a characteristic feature of the superconducting gap, in contrast to the high-temperature EDCs. Moreover, the EDCs for $x = 0.212$ (Fig. 3c, Supplementary Fig. 6c) show the same spectral weight transfer or filling behaviour as with the pristine monolayer FeSe/STO. We conclude that the system evolves into an enhanced superconducting state upon K coating, with the transition point at $x = 0.212$ corresponds exactly to the appearance of the pronounced intensity at $\Gamma$ in the FS.

We now further investigate the enhanced superconductivity of the $x = 0.212$ sample in the momentum space. Figure 3i displays a series of symmetrized EDCs at various $k_F$ points, as indicated in Fig. 3g. The fitting results plotted in a polar representation in

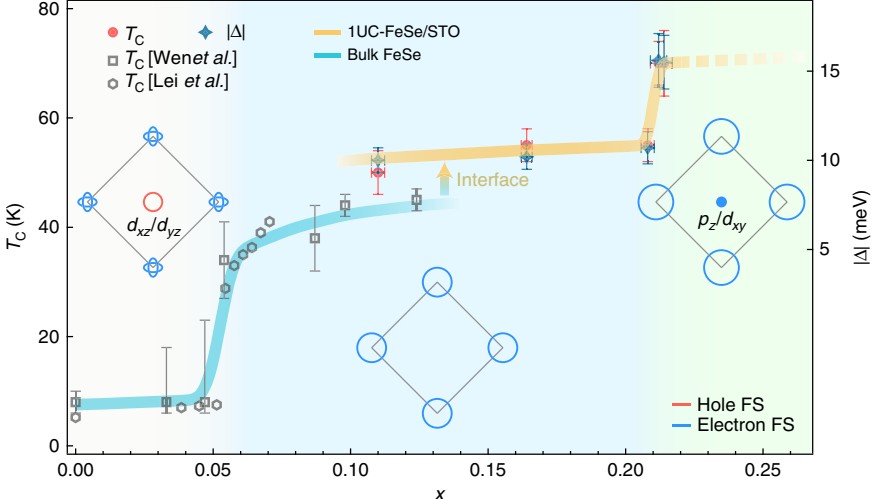

**Figure 4 | Schematic phase diagram of FeSe.** The phase diagram illustrates the evolution of superconductivity and the FS topology. The data at $x = 0.11$ is from our previous study[24]. The $T_c$ data of bulk FeSe, as traced with a cyan curve, is adapted from refs 13,15. The error bars of the superconducting gap include s.d. of the fitting. The error bars of $T_c$ include the measurement uncertainties. We caution that the dashed line is a possible extrapolation result that do not take into account the possible instability of the material for dopings higher than $x = 0.214$.

Fig. 3j show that the superconducting gap around the M point is isotropic within our experimental uncertainties. Since new electronic states appear at $\Gamma$ near $E_F$, we checked the temperature dependence of the EDCs at this point. The EDCs were divided by the FD function and displayed in Fig. 3h. There is also a gap feature here with a size of 15 meV similar to that around M.

**Phase diagram and Lifshitz transitions.** We summarize our results in Fig. 4. The data of bulk FeSe, including K-doped thick FeSe films[13] and liquid-gated FeSe thin flakes[15], are also plotted for comparison, even though they refer to lower doping concentrations than in our study. We note that the electron concentration of FeSe under gating was set based on the gate voltage and the FS evolution of bulk FeSe (refs 13,15). A complex phase diagram differing from the one proposed from ARPES data on bulk FeSe and K-doped thick FeSe films[13] was derived from scanning tunnelling microscopy measurements on multilayer FeSe films[14]. This suggests that the precise doping dependence of the FeSe samples may depend on the exact sample conditions, like the number of layers and the use of a substrate. In any case, we emphasize that all the data from our study correspond to higher dopings than in the other works discussed here[13–15], and thus a direct comparison is not possible.

Between $x = 0.04$ and $x = 0.22$, the FeSe system undergoes two major Lifshitz transitions upon electron doping and the three typical FS topologies are sketched in Fig. 4. Superconductivity is suddenly enhanced at each transition. Based on our knowledge of the first Lifshitz transition, during which the FS pockets around $\Gamma$ vanish, one may expect a suppression of $T_c$ once a FS pocket appears again at the BZ centre. However, our results reveal the precise opposite behaviour. Although we cannot totally rule out a positive influence on $T_c$ of an increase of density-of-states at $E_F$ due to the additional electron pocket, we notice that the $2\Delta/k_B T_c$ ratio is not constant like in the BCS framework, but varies from $\sim 4.5$ to $\sim 5.1$.

**Discussion**

Our observations raise the possible importance of orbital-dependent interactions. Indeed, the $d_{xz}/d_{yz}$ character of the orbitals sinking below $E_F$ across the first Lifshitz transition

(accompanying a jump of $T_c$ (refs 13,15)) is different from the $p_z/d_{xy}$ orbital character emerging at $\Gamma$ across the second Lifshitz transition. Interestingly, in contrast to ARPES measurements on ferropnictide superconductors[22], the outer electron FS pocket at the M point in FeSe/STO, attributed to the $d_{xy}$ orbital, has a larger gap than that of the inner FS pocket according to a recent ARPES study[26].

There is a possible phenomenological explanation to our observation. In bulk FeSe, there is evidence that the $d_{xz}/d_{yz}$ orbitals are strongly linked to the nematic order[27]. Thus, if we assume that the nematic order competes with superconductivity, the absence of the $d_{xz}/d_{yz}$ hole FS may suppress the nematic order and consequently enhance superconductivity, which explains the enhancement at the first Lifshitz transition. In the extended $s$-wave or $s\pm$ pairing scenario, we propose that the emergence of the $d_{xy}/p_z$ electron pocket at the second Lifshitz transition may strengthen the $d_{xy}$ intra-orbital pairing, which is consistent with the observation that the gap enhancement is on the pockets attributed to the $d_{xy}$ orbitals. This is in apparent contradiction with the widely spread belief that in ferropnictides the $d_{xz}/d_{yz}$ orbitals play a determining role in the pairing interactions[28–30]. We caution that the different relative band positions and correlation effects of Fe $3d$ orbitals between ferropnictides and ferrochalcogenides may tune the details of the pairing mechanism. Our results call for a microscopic model involving orbital dependence to explain superconductivity and its enhancement in FeSe/STO.

**Methods**

**Growth of thin films.** Monolayer films of FeSe were grown on 0.05 wt% Nb-doped SrTiO$_3$ substrates after degassing for 2 h at 600 °C and then annealing for 12 min at 925 °C. The substrates were kept at 300 °C during the film growth. Fe (99.98%) and Se (99.999%) were co-evaporated from Knudsen cells with a flux ratio of 1:10 (which were measured by a quart crystal balance) and the growth rate of 0.31 UC min$^{-1}$. The growth process was monitored using reflection high-energy electron diffraction. After growth, the FeSe monolayer films were annealed at 350 °C for 20 h (see reflection high-energy electron diffraction image in Supplementary Fig. 7), and subsequently transferred *in situ* into the ARPES chamber.

**ARPES measurements and K deposition.** ARPES measurements were performed at the Institute of Physics, Chinese Academy of Sciences, using a R4000 analyser and a helium discharge lamp, under ultrahigh vacuum better than $3 \times 10^{-11}$ Torr.

The data were recorded with He I$\alpha$ photons ($hv = 21.218$ eV) unless specified otherwise. The energy resolution was set to $\sim 5$ meV for gap measurements and $\sim 10$ meV for the band structure and FS mapping, while the angular resolution was set to $0.2°$. The Fermi level of the samples was determined from a polycrystalline gold reference in electronical contact with the sample. Deposition of the potassium atoms was carried out in the ARPES preparation chamber using a commercial SAES alkali dispenser, during which the samples were kept at low temperature. The detailed sequences are shown in Supplementary Fig. 1.

**Data availability.** The data that support the findings of this study are available from the corresponding author upon reasonable request.

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

## Acknowledgements

We thank Fu-Chun Zhang for useful discussions. This work is supported by grants from the Ministry of Science and Technology of China (2015CB921000, 2015CB921301, 2016YFA0401000, 2016YFA0300300) and the National Natural Science Foundation of China (11574371, 11274362, 1190020, 11334012, 11274381, 11674371).

## Author contributions

Z.-Q.H., X.-L.P. and Y.-J.S. synthesized the samples. X.S., Z.-Q.H., M.-W.Q. and T.Q. performed the ARPES measurements. X.S. analysed the data. X.S., P.R., J.P.H., Y.-J.S. and H.D. wrote the manuscript. J.P.H. and X.-X.W. provided theoretical input. H.D. and Y.-J.S. supervised the project. All authors discussed the paper.

## Additional information

**Competing interests:** The authors declare no competing financial interests.

