## [Peer Review File · Nature Communications]

Reviewers' comments:

Reviewer #2 (Remarks to the Author):

I have carefully read the re-written manuscript and the comments and replies of the authors. To my opinion most of the referee's questions has been properly addresses. To the benefit of the paper some of the authors replies should be included into supplementary materials.

For example, after adding Figure S1, authors might include small paragraph about K deposited on the surface of the FeSe/STO film with all the STM work references as in the following reply:

"Previous STM studies (PRL 116, 157001 (2016), PRB 93, 020507(R) (2016), Nano Lett. 16, 1969 (2016)) reported that the K atoms stay on the surface of the film. Parts (could be all with very low coverage) of them are ionized, as supported by the observation of, the K atom clusters with increasing coverage ($>0.2\text{ML}$) in the STM image. Unfortunately, we are not equipped with a low-temperature STM with atomic resolution. However, our ARPES (a sophisticated version of photoemission compared to XPS) data clearly show a chemical potential shift, thus indicating electron-doping in our experiment. The extra electrons can only come from the ionization of K."

Another advantage will be if authors add estimate for the K coverage in their experiment in number of monolayers of potassium deposited in order to make a comparison with STM results.

As for the Figure S3, authors in their reply mentioned that,

"We are dealing with monolayers of FeSe. In addition to the doping effect, the deposited K atoms perturb the structure of FeSe. The precise bond lengths and angles are thus different for the Se atoms above and below the Fe planes. This distortion of the crystal structure naturally leads to a distortion of the band structure, as determined theoretically (Supplementary material, Fig. S3)." Noting in the corresponding figure caption that internal atomic positions are fully relaxed authors might include a table with atomic positions. Because crystal structure of monolayer FeSe/STO has been not determined and agreed, this theoretical values would be very useful for experimentalists.

And my last suggestion concerns Figure 2 with ARPES energy dispersions of holes (a) and electrons (b).

According to the figure captions, the dashed red curves are parabolic fits to the band dispersions. It will be very helpful to extract corresponding effective masses and plot then as a function of doping for comparison with K-doped bulk FeSe work by Wen et al. [Ref 9] where renormalisation increases despite decrease of T_c with overdoping.

Few major comments:

Sentence about density of states (DOS) at Fermi level [lines 131-135] are somehow confusing, stating that possible increase of DOS due to additional electron pocket is in contrast to increase of DOS in BCS theory. Does this sentence actually mean that this is not BCS-type superconductivity because of $\{2\Delta/T_c\}$ is not constant? Please clarify on this.

And at last, I strongly disagree with the last sentence in the Discussion section:

"Orbital-dependent AFM interactions must be required in order to understand our results, even qualitatively."

Nothing in the paper points towards antiferromagnetic nature of the interaction. If you insist on this, please put corresponding arguments before.

Reviewer #3 (Remarks to the Author):

The authors have satisfactorily responded to all the questions (including unimportant ones) raised by Reviewer #1, and also those by Reviewer #2. The discovery that the T_c enhancement and the Lifshitz transition occur simultaneously is very interesting and would be important. I recommend the paper to be published in Nature Communications if the authors address the following points.

1. One of a few weak points of the paper is the claim that the d_{xy} orbitals composing the electron pocket at the Gamma point plays the leading role in superconductivity. This may not be supported by a majority of theoretical studies on Fe pnictides, according to which the d_{yz}/d_{zx} orbitals are responsible for superconductivity and the d_{xy} orbitals are too strongly correlated to support superconductivity. The authors should clarify this point by referring to relevant theoretical work.

2. Another concern is that, in Fig.4, the horizontal dashed line extrapolating the T_c and gap values to $x > 0.214$ is misleading, because there is a high probability that the material becomes unstable above $x = 0.214$ as in many other superconductors whose T_c reaches the maximum at the stability limit of the material.

Reviewers' comments:

Reviewer #2 (Remarks to the Author):

We thank Reviewer #2 for his careful re-review of our manuscript.

I have carefully read the re-written manuscript and the comments and replies of the authors.

To my opinion most of the referee's questions has been properly addresses. To the benefit of the paper some of the authors replies should be included into supplementary materials.

For example, after adding Figure S1, authors might include small paragraph about K deposited on the surface of the FeSe/STO film with all the STM work references as in the following reply:

“Previous STM studies (PRL 116, 157001 (2016), PRB 93, 020507(R) (2016), Nano Lett. 16, 1969 (2016)) reported that the K atoms stay on the surface of the film. Parts (could be all with very low coverage) of them are ionized, as supported by the observation of, the K atom clusters with increasing coverage ($>0.2\text{ML}$) in the STM image. Unfortunately, we are not equipped with a low-temperature STM with atomic resolution. However, our ARPES (a sophisticated version of photoemission compared to XPS) data clearly show a chemical potential shift, thus indicating electron-doping in our experiment. The extra electrons can only come from the ionization of K.”

Another advantage will be if authors add estimate for the K coverage in their experiment in number of monolayers of potassium deposited in order to make a comparison with STM results.

Reply:

We added related descriptions to the supplementary materials following the reviewer's suggestion.

Regarding the K deposition onto a sample, STM measures the real coverage and ARPES measures the real chemical doping. These two can be related or compared if all the deposited K atoms are ionized. This is not the case in K doped FeSe as we showed in Figure S1.

As for the Figure S3, authors in their reply mentioned that,

“We are dealing with monolayers of FeSe. In addition to the doping effect, the deposited K atoms perturb the structure of FeSe. The precise bond lengths and angles are thus different for the Se atoms above and below the Fe planes. This distortion of the crystal structure naturally leads to a distortion of the band structure, as determined theoretically (Supplementary material, Fig. S3).”

Noting in the corresponding figure caption that internal atomic positions are fully relaxed authors might include a table with atomic positions. Because crystal structure of monolayer FeSe/STO has been not determined and agreed, this theoretical values would be very useful for experimentalists.

Reply:

We added the table of atomic positions in the calculations to the Fig. S3.

And my last suggestion concerns Figure 2 with ARPES energy dispersions of holes (a) and electrons (b).

According to the figure captions, the dashed red curves are parabolic fits to the band dispersions. It will be very helpful to extract corresponding effective masses and plot then as a function of doping for comparison with K-doped bulk FeSe work by Wen *et al.* [Ref 9] where renormalisation increases despite decrease of T_c with overdoping.

Reply:

The extracted effective masses of the hole-like band at Γ and electron-like band at M are plotted in the following figure. Firstly, the result is consistent with the work by Wen *et al.* [Ref 13 in the revised manuscript]. Secondly, the effective mass behaves in the opposite ways for these two bands. Thirdly, we have further evidences (in a new manuscript that in prepare) to show that the effective mass is not directly linked to the superconductivity in this system.

Few major comments:

Sentence about density of states (DOS) at Fermi level [lines 131-135] are somehow confusing, stating that possible increase of DOS due to additional electron pocket is in contrast to increase of DOS in BCS theory. Does this sentence actually mean that this is not BCS-type superconductivity because of $\{2\Delta/T_c\}$ is not constant? Please clarify on this.

Reply:

In the BCS theory, T_c is proportional to the DOS at E_F , but the $2\Delta/k_B T_c$ ratio does not vary. We are convinced, as most of the community, that the Fe-based superconductors are host to an unconventional pairing mechanism that is different from the BCS theory, which is a mean-

field theory. We do not attribute the sudden enhancement of superconductivity to the increase of DOS following the BCS theory. In fact, by showing that the $2\Delta/k_B T_c$ ratio varies, we show inconsistency with the BCS theory. We have reworded this section in the manuscript in order to avoid confusion.

And at last, I strongly disagree with the last sentence in the Discussion section:

“Orbital-dependent AFM interactions must be required in order to understand our results, even qualitatively.”

Nothing in the paper points towards antiferromagnetic nature of the interaction. If you insist on this, please put corresponding arguments before.

Reply:

We removed this sentence in the updated manuscript.

Reviewer #3 (Remarks to the Author):

We thank Reviewer #3 for his careful review of our manuscript.

The authors have satisfactorily responded to all the questions (including unimportant ones) raised by Reviewer #1, and also those by Reviewer #2. The discovery that the T_c enhancement and the Lifshitz transition occur simultaneously is very interesting and would be important. I recommend the paper to be published in Nature Communications if the authors address the following points.

Reply:

We thank Reviewer #3 for recognizing the importance of our work.

1. One of a few weak points of the paper is the claim that the d_{xy} orbitals composing the electron pocket at the Gamma point plays the leading role in superconductivity. This may not be supported by a majority of theoretical studies on Fe pnictides, according to which the d_{yz}/d_{zx} orbitals are responsible for superconductivity and the d_{xy} orbitals are too strongly correlated to support superconductivity. The authors should clarify this point by referring to relevant theoretical work.

Reply:

In the new manuscript we cited relevant studies on Fe pnictides and made further discussions about our results. We raise an open question regarding the contrast of dominating orbitals in FeSe and Fe pnictides, which is also supported by the work of Ref. 28. In any case, the point

we are trying to make is that the orbital dependence should be taken into account to explain superconductivity and its enhancement in FeSe/STO.

2. Another concern is that, in Fig.4, the horizontal dashed line extrapolating the T_c and gap values to $x > 0.214$ is misleading, because there is a high probability that the material becomes unstable above $x = 0.214$ as in many other superconductors whose T_c reaches the maximum at the stability limit of the material.

Reply:

This concern is reasonable. We updated Figure 4 to make the dashed line more inconspicuous, and explicitly pointed out the possibility suggested by the reviewer in the figure caption.

*****END*****